# Multiplex Screening for Interacting Compounds in Paediatric Acute Myeloid Leukaemia

**DOI:** 10.3390/ijms221810163

**Published:** 2021-09-21

**Authors:** Lauren V. Cairns, Katrina M. Lappin, Alexander Mutch, Ahlam Ali, Kyle B. Matchett, Ken I. Mills

**Affiliations:** 1Patrick G. Johnston Centre for Cancer Research, Queen’s University Belfast, 97 Lisburn Road, Belfast BT9 7AE, UK; K.Lappin@qub.ac.uk (K.M.L.); a.ali@qub.ac.uk (A.A.); k.mills@qub.ac.uk (K.I.M.); 2Northern Ireland Centre for Stratified Medicine, School of Biomedical Sciences, Ulster University, C-TRIC, Altnagelvin Hospital Campus, Glenshane Road, Derry/Londonderry BT47 6SB, UK; amutch02@qub.ac.uk (A.M.); k.matchett@ulster.ac.uk (K.B.M.)

**Keywords:** acute myeloid leukaemia, paediatric, drug repurposing, multiplex screening

## Abstract

Paediatric acute myeloid leukaemia (AML) is a heterogeneous disease characterised by the malignant transformation of myeloid precursor cells with impaired differentiation. Standard therapy for paediatric AML has remained largely unchanged for over four decades and, combined with inadequate understanding of the biology of paediatric AML, has limited the progress of targeted therapies in this cohort. In recent years, the search for novel targets for the treatment of paediatric AML has accelerated in parallel with advanced genomic technologies which explore the mutational and transcriptional landscape of this disease. Exploiting the large combinatorial space of existing drugs provides an untapped resource for the identification of potential combination therapies for the treatment of paediatric AML. We have previously designed a multiplex screening strategy known as Multiplex Screening for Interacting Compounds in AML (MuSICAL); using an algorithm designed in-house, we screened all pairings of 384 FDA-approved compounds in less than 4000 wells by pooling drugs into 10 compounds per well. This approach maximised the probability of identifying new compound combinations with therapeutic potential while minimising cost, replication and redundancy. This screening strategy identified the triple combination of glimepiride, a sulfonylurea; pancuronium dibromide, a neuromuscular blocking agent; and vinblastine sulfate, a vinca alkaloid, as a potential therapy for paediatric AML. We envision that this approach can be used for a variety of disease-relevant screens allowing the efficient repurposing of drugs that can be rapidly moved into the clinic.

## 1. Introduction

Acute myeloid leukaemia (AML) is a heterogeneous disease characterised by the malignant transformation of myeloid precursor cells with impaired differentiation [1,2]. AML is often considered as a disease of the elderly with incidence rising markedly with age; the median age at diagnosis is approximately 70 years old [3]. Childhood AML accounts for just 20% of all acute leukaemia cases despite being responsible for more deaths than the more common acute lymphocytic leukaemia (ALL) [4]. Currently, the overall survival (OS) rates for paediatric AML range between 60 and 70%, which is a significant improvement from the 40% 5-year survival rates observed in the early 1980s. This can be attributed to more intense chemotherapeutic regimens and improved supportive care [5,6]. Despite these significant improvements, relapse and refractory AML remains a leading cause of mortality in children [7].

Until recently, our understanding of paediatric AML has been influenced by genomic data collated from adult cases. However, high-throughput sequencing of paediatric samples has revealed that paediatric AML is biologically different from adult AML [8,9]. The Therapeutically Applicable Research to Generate Effective Treatments (TARGET) AML initiative provided a comprehensive characterisation of the mutational, transcriptional and epigenetic landscapes of paediatric AML [10]. Frequently mutated genes among adult cases included *DNMT3A* and *NPM1*, while these mutations were relatively rare in paediatric cases (<2%) [9,10,11]. Converse to this, mutations in *KIT, KRAS* and *NRAS* were significantly associated with paediatric AML in comparison to adults [9,10,11]. The 7 + 3 regimen is the most widely used therapeutic approach for the treatment of both paediatric and adult AML, which involves a 7-day continuous infusion of cytarabine supplemented with an anthracycline during days 1–3 [12]. This standard of care has remained largely unchanged for over four decades [13]. Furthermore, these therapies have demonstrated multiple long-term and late effects for survivors of childhood AML, including impaired cognitive effects and physical development and infertility [14,15]. Inadequate understanding of the biology of paediatric AML has limited the progress of targeted therapies in this cohort. As evidenced by the TARGET study, improving our understanding of the genomic alterations associated with paediatric AML can improve patient stratification and aid in the development of targeted therapies [10].

Combination therapies have long been a preferable and superior approach for the treatment of paediatric leukaemia. Moon et al. (1965) first highlighted the effectiveness of combining antileukaemic agents to improve the duration and frequency of complete remission in children [16]. Combination therapies have the potential to work in a synergistic or additive manner and thus allow a lower therapeutic dose to be achieved, minimising potential long-term effects [17]. Clinical resistance remains a challenge in the treatment of all cancer types, and combination therapies have the potential to overcome adaptive resistance by targeting multiple disease drivers [18]. Identifying effective combination therapies is characterised by multiple challenges and limitations, particularly due to the lengthy drug development process [19]. In 2020, less than 4% of novel drug approvals by the U.S. Food and Drug Administration (FDA) demonstrated efficacy for haematological malignancies [20], highlighting the potential for drug repurposing. Drug repurposing offers an alternative strategy, which involves finding new applications for existing compounds, providing a more time- and cost-effective approach [19]. The literature highlights multiple cases of drug repurposing in blood cancers [21,22,23], the most established example being thalidomide, which was originally marketed as an antiemetic for the treatment of nausea in pregnant women. It was later shown to be effective in the treatment of multiple myeloma [24].

In this study, we demonstrate the utility of a novel multiplex screening approach [8] to identify potential repurposed combination therapies for the treatment of paediatric AML. We identified the combination of glimepiride, a sulfonylurea used in the management of type 2 diabetes mellitus [25]; pancuronium dibromide, a neuromuscular blocking agent [26]; and vinblastine sulfate, a vinca-alkaloid anticancer agent [27], as a potential therapy for paediatric AML.

## 2. Results

### 2.1. Multiplex Screening Using an All-Pairs Testing Algorithm to Identify Combination Therapies In Vitro

To test all pairings of 384 compounds, 73,536 (=384 × (384 − 1)/2) individual wells would be required using standard approaches. As described previously, an all-pairs testing algorithm, designed in-house, was used to group compounds into pools of 10 compounds per well, allowing all possible pairwise combinations of 384 drugs to be accommodated in 3878 wells [8]. The 3878 10-compound combinations were created using the Echo liquid handling technology (Labcyte); two paediatric AML cell lines (CMK and MV4-11) were treated with the combinations using the optimum dose per drug, and cell viability was assessed at 48 and 72 h. The heatmaps in Figure 1A highlight the Z-score analysis of each of these individual combinations at both time points across each cell line. Z-score correlates to loss of membrane integrity, and a higher Z-score (red) indicates induction of cell death. A Z-score greater than 2 (>2) was identified as representing a significant decrease in cell viability. Interestingly, more than 94% of combinations did not cause significant induction of cell death despite the presence of 10 drugs per well. For the MV4-11 cell line, 5.3% (205/3878) of combinations demonstrated Z-score > 2; in contrast, only 3.7% (143/3878) of combinations in the CMK cell line demonstrated a significant decrease in cell viability. Concurrently with the combination screen, cell lines were treated with each of the 384 compounds as single agents, and cell viability was assessed at 48 and 72 h (Figure 1B). Each of the cell lines demonstrated variability in response to the single agents; of the 384 compounds, only 3 agents demonstrated a common effect across both cell lines—bortezomib, gemcitabine and topotecan. The single-agent screen was used to refine the effectiveness of each of the successful combinations. For a combination to be considered a ‘hit’ it must have met the following criteria: (1) the 10-compound combination demonstrated a Z-score > 2 and (2) each of the compounds in the combination exhibited a Z-score < 2 as a single agent. In the CMK cell line, of the 143 combinations that demonstrated a Z-score > 2, only 43 of these combinations were considered ‘hits’ (30.2% or 1.1% of total combinations). Similarly, for the MV4-11 cell line, 195 combinations showed a Z-score > 2, but only 30 of these combinations were considered ‘hits’ (15.4% or 0.8%) (Figure 1C). Interestingly nine (0.24%) ‘hit’ combinations had a common effect across both cell lines (Figure 1C). These nine overlapping combinations are represented graphically in Appendix A. Mycophenolate mofetil, an inhibitor of inosine monophosphate dehydrogenase (IMPDH) [28], was the most frequently occurring drug across these overlapping combinations, appearing in four out of nine wells (Appendix A).

### 2.2. Deconvolution of ‘Hit’ Wells to Investigate Double and Triple Combinations as Potential Therapies for the Treatment of Paediatric AML

As the main focus of this work was to identify a robust drug combination, there were nine combinations across both cell lines that demonstrated an increase in RFU indicating induction of cell death. Of particular interest was Plate 13 Well 115 as the 10 compounds within this well (glipizide, prazosin, pancuronium dibromide, carbachol, escitalopram oxalate, simvastatin, tolazamide, amifostine, vinblastine sulfate and glimepiride) had minimal or no effect as single agents on the viability of either cell line. In contrast, as a combination, the drug pool resulted in a 3-fold increase in Z-score in both cell lines; this combination was shown to demonstrate the highest Z-score for both cell lines (Figure 2A). To identify a potential synergistic combination, the 10-compound combination was deconvoluted into 45 pairwise combinations and 120 triple combinations. The viability of these combinations was tested over a period of 72 h at a final concentration of 0.1 µM with the exception of vinblastine sulfate, which had a final concentration of 0.001 µM as this concentration was more clinically achievable. Initial investigations looked at the Z-score of each of the 10 compounds as single agents; all of the compounds demonstrate a Z-score < 2 in both CMK (Figure 2A(i)) and MV4-11 (Figure 2A(ii)) cell lines, while the 10-compound combination demonstrates a Z-score > 2. Deconvolution into 45 pairwise combinations demonstrated variability in each cell; the pairwise combination with the highest Z-score (increased loss of membrane integrity) for the CMK cell line was escitalopram oxalate, a selective serotonin reuptake inhibitor (SSRI) [29], and amifostine, a cytoprotective adjuvant [30] (Z-score = 2.19) (Figure 2B(i)), while the highest-scoring pairwise combination for the MV4-11 cell line was prazosin, an alpha-1 blocker [31], and simvastatin, a lipid-lowering medication [32] (Z-score = 2.04) (Figure 2B(ii)). Further to this, all 120 triple combinations for this 10-compound combination were investigated, and similar to the pairwise combinations, each cell line demonstrated variability. For the CMK cell line, one triple combination demonstrated a Z-score > 2; this was escitalopram oxalate in combination with two sulfonylureas [33,34], tolazamide and glimepiride (Z-score = 2.06). Two triple combinations that generated a Z-score > 2 were observed in MV4-11 cells, which included a sulfonylurea (either tolazamide or glimepiride) in combination with pancuronium dibromide, a neuromuscular relaxant, and vinblastine sulfate, a chemotherapeutic agent (Z-score = 2.13 and 2.30, respectively) (Figure 2C). These pairwise and triple combinations were taken forward for validation and further analysis.

### 2.3. Validation of Double and Triple Combinations Identified in the Deconvolution of Plate 13 Well 115

Deconvolution of the 10-compound combination into 45 pairwise combinations and 120 triple combinations identified two pairwise combinations and three triple combinations across both cell lines (Table 1). To validate these findings, we first used CellTox Green Express Cytotoxicity assay, and cells were treated over a 72 h period with each of the compounds as single agents alongside the appropriate combination. RFU values were normalised to DMSO controls, and a Z-score was calculated based on the mean and standard deviation of the data; Z-score correlates to loss of membrane integrity. Only one of the five combinations tested demonstrated a statistically significant increase in Z-score across both cell lines; this was glimepiride, pancuronium dibromide and vinblastine sulfate. This combination resulted in an almost 2-fold increase in Z-score when compared to the single agents (Figure 3A(iv)). Further to this, the combination of tolazamide, pancuronium dibromide and vinblastine sulfate was also shown to demonstrate a statistically significant increase in Z-score in the MV4-11 cell line (Figure 3B(v)); for the CMK cell line, this combination was also shown to demonstrate an increase in Z-score compared to the single agents, but this was not shown to be significant (Z-score = 2.05) (Figure 3B(v)). To further investigate the effects of these combinations on cell viability using CellTiter-Glo luminescence assay, as previously described, cells were treated over a 72 h period with each of the compounds as single agents alongside the appropriate combination. Relative luminescence values (RLU) were normalised to DMSO controls, and cell viability was subsequently calculated. Like the CellTox Green analysis, the combinations of glimepiride, pancuronium dibromide and vinblastine sulfate (Figure 3B(iv)) and tolazamide, pancuronium dibromide and vinblastine sulfate (Figure 3B(v)) demonstrated a statistically significant decrease in cell viability across both cell lines. Following the successful identification of these novel combinations, we aimed to investigate the nature of their interaction by providing a quantitative definition for a synergistic relationship. Combination index (CI) values were calculated for these combinations using the Chou and Talalay method [35]. Dose–response curves for each of the single agents were used to identify the fraction affected at multiple concentrations (Appendix A), and this was compared to the fraction affected of the combinations in the CompuSyn software (ComboSyn Inc.). CI value >1 is antagonistic, CI value = 1 is additive and CI value < 1 is synergistic. Both the CMK (Figure 4A(i)) and MV4-11 (Figure 4A(ii)) cell lines demonstrated a synergistic relationship when treated with the combination of glimepiride, pancuronium dibromide and vinblastine sulfate, with CI values < 1, except for one concentration (glimepiride (1 µM), pancuronium dibromide (1 µM) and vinblastine sulfate (0.001 µM)) in the CMK cell line that demonstrated an additive effect (CI = 1.14521) (Table 2). Furthermore, we investigated the relationship of tolazamide, pancuronium dibromide and vinblastine sulfate; despite demonstrating a statistically significant decrease in cell viability, this combination was shown to be an antagonist in both CMK (Figure 4B(i)) and MV4-11 (Figure 4B(ii)) cell lines, with CI values > 1 (Table 3). Moving forward, we aimed to investigate the mechanism of cell death associated with the combination of glimepiride, pancuronium dibromide and vinblastine sulfate.

### 2.4. The Combination of Glimepiride, Pancuronium Dibromide and Vinblastine Sulfate Induces Apoptotic Cell Death in Paediatric AML

CMK (Figure 5A(i)) and MV4-11 (Figure 5A(ii)) cell lines were treated with glimepiride, pancuronium dibromide and vinblastine sulfate as single agents and as a combination; following a 72 h incubation, protein was extracted from the cells and Western blot analysis was performed. Both cell lines demonstrated a statistically significant increase in PARP cleavage and caspase-3 cleavage following treatment with the triple combination of glimepiride, pancuronium dibromide and vinblastine sulfate as compared to the single agents (Figure 5A). Subsequently, both cell lines also exhibited a marked decrease in procaspase-3 expression following treatment with this triple combination (Figure 5A). Densitometry analysis was performed by comparing the relative density of each sample following normalisation to GAPDH loading control (Appendix A). Enhanced expression of cleaved caspase-3 alongside cleaved PARP is indicative of apoptotic cell death. To investigate apoptotic cell death as a mechanism of action, both CMK and MV4-11 cell lines were stained with annexin V/PI conjugates following 72 h treatment with glimepiride, pancuronium dibromide and vinblastine sulfate as single agents and as a combination, and cells were analysed using flow cytometry. Flow cytometry analysis revealed that both CMK (Figure 5B(i)) and MV4-11 (Figure 5B(ii)) cell lines demonstrated a statistically significant increase in annexin V-positive cells in the combination as compared to the single agents. Flow cytometry plots demonstrate the shift in the cell population towards annexin V positivity (right side) following treatment with this triple combination (Figure 5C). The combination of glimepiride, pancuronium dibromide and vinblastine sulfate can be deconvoluted into three pairwise combinations, namely glimepiride and pancuronium dibromide, glimepiride and vinblastine sulfate and pancuronium dibromide and vinblastine sulfate. The effectiveness of these pairwise combinations was investigated across both CMK and MV4-11 cell lines; CellTiter-Glo luminescence assay demonstrated no significant changes in cell viability for each of the pairwise combinations when compared to the single agents (Appendix A). Further to this, these pairwise combinations demonstrated no significant changes in annexin V positivity (Appendix A). These investigations further confirm the effectiveness of this triple combination. 

### 2.5. The Combination of Glimepiride, Pancuronium Dibromide and Vinblastine Sulfate Demonstrates Reduced Cell Viability across Multiple Paediatric Cell Lines

The novel triple combination of glimepiride, pancuronium dibromide and vinblastine sulfate was investigated to establish the effectiveness across a wider panel of paediatric AML cell lines (MOLM-13, Kasumi-1, CMS, MOLM-14, THP-1, CMK and MV4-11). Interestingly, the data indicate that five out of the seven (MOLM-13, Kasumi-1, MOLM-14, CMK and MV4-11) cell lines demonstrated a synergistic response (CI value = 0.96967, 0.90311, 0.90859, 0.89559 and 0.93907, respectively) when treated with the triple combination of glimepiride, pancuronium dibromide and vinblastine sulfate (Figure 6). In contrast, CMS and THP-1 cell lines demonstrate an antagonistic relationship following treatment with this triple combination (CI values = 3.88 and 2.389, respectively) (Figure 6). Further investigation indicated that the MOLM-13, Kasumi-1 and CMK cell lines demonstrated a statistically significant increase in annexin V-positive cells when treated with the combination of glimepiride, pancuronium dibromide and vinblastine sulfate in comparison to the single agents. This was accompanied by enhanced expression of cleaved PARP and cleaved caspase-3 and reduced expression of procaspase 3. In contrast, the CMK cell line did not demonstrate any increase in annexin V-positive cells when treated with this triple combination. Western blot analysis also demonstrated no change in the expression of cleaved PARP, cleaved caspase-3 and procaspase-3 (Appendix A).

## 3. Discussion

The current treatment of paediatric AML has reached a therapeutic plateau; further intensification of chemotherapy is limited by severe toxicity with minimal improvements in OS [36]. Alongside this, many paediatric patients experience long-lasting complications of this intense chemotherapy regime, impacting their quality of life [37]. Even with this intensive treatment, up to 30% of paediatric patients experience relapse [37]. Targeted therapies have the potential to improve antileukaemic efficacy while minimising treatment-related morbidity and mortality. Enhanced understanding of the molecular landscape of paediatric AML has led to improved risk stratification and provided new therapeutic targets [38]. These include gene mutations such as *FLT-3* and *KIT*, deregulated signalling pathways such as mitogen-activated protein kinases (MAPKs) and fusion proteins such as *PML-RARA* [38]. Rearrangement of the *MLL* gene is one of the most common genetic events to occur in paediatric AML; *MLL* rearranges with more than 80 different partner genes, and the resulting fusion proteins deregulate expression of *MLL* target genes [39]. *MLL*-rearranged leukaemic cells are dependent on the serine/threonine kinase glycogen synthase kinase-3 (GSK3), prompting the investigation of GSK3 inhibitors as potential therapies in paediatric AML [39]. A preclinical murine model of *MLL* leukaemia demonstrated significant prolongation of survival following treatment with lithium carbonate, a known GSK3 inhibitor [40]. Despite this, only one targeted therapy (Mylotarg) has been FDA-approved for the treatment of paediatric AML since 2017 [41].

Drug repurposing as an alternative method offers multiple advantages; compounds already in clinical use will typically have a known safety and toxicity profile, leading to accelerated timelines and reduced costs [19]. The unaffordable prices of traditionally developed compounds combined with an unmet clinical need to improve OS in AML have encouraged the exploration of repurposing as a feasible strategy [42]. Several examples of drug repurposing in AML have been identified in the literature [43,44]; most often, these arise serendipitously through observed side effects, as evidenced by thalidomide [20]. More recently, compounds have been identified through the application of high-throughput screening (HTS) [45]. Drenberg et al. (2019) highlighted the potential to improve treatment success using a repurposing strategy; they evaluated the antileukaemic activity of over 7000 FDA-approved compounds across a panel of paediatric AML cell lines [45]. This screening strategy demonstrated the utility of the nucleoside analogue gemcitabine and the JAK inhibitor cabazitaxel as potential therapies for paediatric AML [45]. With combination therapy being the optimal approach for the treatment of paediatric AML, our study set out to investigate the large combinatorial space of repurposed therapies with the aim of identifying combination therapies that demonstrate efficacy in this cohort.

Previously published data from our group detailed the efficacy of a novel screening strategy to identify potential combination therapies for the treatment of paediatric AML [8]. Using an algorithm designed in-house, this screening strategy investigated all possible pairwise combinations of a targeted library of 80 apoptosis-inducing agents by grouping compounds into pools of 10 compounds per well; this approach maximises the probability of identifying new compound combinations with therapeutic potential while minimising replication and redundancy [8].

Using this screening strategy, in the current study, we investigated all combinations of 384 FDA-approved compounds across two paediatric AML cell lines (CMK and MV4-11). Using a standard approach, this would require over 73,000 individual wells; however, using our screening strategy, we can reduce this to 3878 individual wells. For the most part, each of the cell lines demonstrated variability in response to these compound combinations, highlighting the heterogeneity of paediatric AML and the need for a personalised medicine approach [46,47]. We focused on compound combinations that demonstrated similar responses in both cell lines with the aim of identifying a robust combination. Deconvolution of Plate 13 Well 115 into double and triple combinations revealed the novel triple combination of glimepiride, pancuronium dibromide and vinblastine sulfate as a potential therapy for paediatric AML. It is our understanding that this combination has not been reported previously.

Glimepiride is a sulfonylurea used in the treatment of type 2 diabetes mellitus; unlike similar medications such as metformin, glimepiride promotes the secretion of insulin and increases the expression of IGF, having the potential to promote tumorigenesis as a single agent through activation of mitogen-activated protein kinase and PI3K–Akt–mTOR pathways [48]. The association between sulfonylureas and cancer risk has been extensively investigated in the literature; a 2012 study by Chang et al. indicated that significantly increased risks were found for first- and second-generation sulfonylureas (OR, 1.08; 95% CI, 1.01–1.15), but not for glimepiride (OR, 1.00; 95% CI, 0.93–1.08) [49]. Studies have indicated that glimepiride has a lower ability to stimulate insulin production in comparison to conventional sulfonylureas such as tolazamide [50]. The potential anticancer effects of sulfonylureas have also been explored in the literature; impaired angiogenesis and vasculogenesis are known to occur through inhibition of VEGF alongside sensitivity to chemotherapy through inhibition of ABC transporters [51]. Most of the data available are surrounding glibenclamide, with no clear evidence of anticancer effects following treatment with glimepiride.

Pancuronium is a non-depolarising neuromuscular blocker used as a muscle relaxant during anaesthesia; it was first synthesised in 1964, and it acts as a competitive inhibitor at nicotinic acetylcholine (ACh) receptors [52]. There is no literary evidence of an association between pancuronium dibromide and cancer risks, nor is there any indication that pancuronium dibromide exhibits any anticancer effects.

Finally, vinblastine sulfate, this compound is a first-generation vinca alkaloid that causes cell-cycle arrest by inhibiting the polymerisation of tubulin, which is necessary for spindle formation during mitosis [53]. Vinblastine was originally isolated from the Madagascar periwinkle *Catharanthus roseus* (formerly *Vinca rosea*) over 50 years ago [54]. The anticancer activity of this species was discovered during a diabetes study; researchers noted a significant reduction in white cell count and destruction of bone marrow in rats following treatment with *C. roseus* extract [54]. Vinblastine sulfate was FDA-approved in 1965 under the brand name Velban [55]. Early clinical trials highlighted the efficacy of vinblastine in the treatment of Hodgkin’s lymphoma and choriocarcinoma [56]. Further investigation demonstrated response to vinblastine in both bladder and breast cancers. Despite demonstrating significant anticancer activity in lymphomas, vinblastine sulfate is not typically used in the treatment of AML. Studies have indicated scope for the use of vinblastine sulfate in combination with other agents as potential treatments for leukaemias. Eastman et al. demonstrated the utility of the MEK inhibitor PD98059 to sensitise ML-1 leukaemia cells to vinblastine-mediated apoptosis [57]. Further to this, a 2011 study indicated that treatment with vinblastine sulfate sensitised chronic lymphocytic leukaemia (CLL) cells to the cyclin-dependent kinase inhibitors flavopiridol and dinaciclib [58]. Vincristine is a closely related compound to vinblastine, which is primarily used in the treatment of acute lymphoblastic leukaemias (ALLs) and lymphomas; vincristine sulfate was frequently observed in the overlapping successful hits from our combination screen.

Identification of the triple combination of glimepiride, pancuronium dibromide and vinblastine sulfate demonstrates the utility of this screening approach to establish personalised treatment strategies for paediatric AML. The mutational landscape of paediatric AML contributes to a complex therapeutic landscape, and variability in response to this combination across multiple cell lines highlights the potential variability amongst patients. The complexity of this interaction requires further investigation to elucidate the exact mechanism of action.

## 4. Materials and Methods

### 4.1. Cell Lines

MV4-11 and CMK cell lines were used during this study to represent the heterogeneity of paediatric AML. The MV4-11 cell line was established from the blast cells of a 10-year-old male that presented with biphenotypic B-myelomonocytic leukaemia at diagnosis (AML FAB M5) [59]. The CMK cell line was established from the peripheral blood of a 10-month-old boy with Down’s syndrome and acute megakaryocytic leukaemia (AML M7) at relapse in 1985. The MOLM-13, Kasumi-1, MOLM-14, THP-1 and CMK cell lines were used for validation experiments. All cell lines were obtained from the Deutsche Sammlung von Mikroorganismen und Zellkulturen (DSMZ, Germany, EU) with the exception of the CMS cell line, which was provided courtesy of Dr Yubin Ge (Detroit, MI, USA) [60]. Each cell line was maintained in Roswell Park Memorial Institute (RPMI) 1640 (Thermo Fisher Scientific, Paisley, UK) supplemented with either 10% foetal bovine serum (FBS; Thermo Fisher Scientific, Paisley, UK) (MV4-11) or 20% FBS (CMK) and 100 µg/mL penicillin–streptomycin (Thermo Fisher Scientific, Paisley, UK). All cell lines were cultured in a humidified incubator at 37 °C supplemented with 5% carbon dioxide.

### 4.2. Combination Screen Algorithm

An all-pairs testing algorithm, as described by Lappin et al. (2020) [8], was used to group drugs into pools of 10 compounds per well. All pairings of 384 FDA-approved compounds were screened; using this algorithm, the number of wells was reduced from ~73,000 to 3878.

### 4.3. Compound Screening

The Enzo Life Sciences SCREEN-WELL FDA-approved drug library V2 was the basis for this multiplex screen. The compound screening assay was carried out in 384-well black optical bottom plates (Nunc, Science Warehouse Limited). Cell lines (MV4-11 and CMK) were seeded at a density of 2 × 10^4^ cells per well. Compounds were diluted with blank media and added to cells to generate a final drug concentration of 0.1 µM. Cells treated with 0.1% DMSO were used as a vehicle control. Cells were treated for 72 h in a humidified incubator at 37 °C supplemented with 5% CO_2_. Cell viability was assessed at 48 and 72 h using CellTox Green Express Cytotoxicity Assay (Promega, Southhampton, UK), which was added to the cell suspension at a 1:1000 dilution prior to seeding. Relative fluorescence unit (RFU) (Ex: 485 nm, Em: 520 nm) was measured using a Synergy HTX Multi-mode Microplate reader (Biotek, Winooski, VT, USA).

### 4.4. Z-Score Calculation

The RFU value for each well was normalised to the average RFU value for the DMSO controls on that plate. The mean and standard deviation were calculated for the normalised RFU values, and a Z-score was assigned using the formula below. Combinations that demonstrated a Z-score > 2 were taken forward for further investigation.
(1)$$Z=\frac{\left(x-µ\right)}{\sigma }$$

x = normalised RFU values;

µ = mean;

σ = standard deviation.

### 4.5. Deconvolution

A combination was considered successful when the individual agents demonstrated minimal single-drug activity (Z-score < 2) but when cell death was observed in combinations of 10 (Z-score > 2). In order to identify clinically feasible combinations from hit wells, the 10-compound combination was deconvoluted into 45 pairwise combinations and 120 triple combinations.

### 4.6. Cell Viability Assay

Cell viability was assessed using CellTiter-Glo (Promega, Southhampton, UK) luminescent assay. Briefly, cells were seeded at 2 × 10^5^/mL in a 12-well plate, appropriate drug treatments were added and the plates were incubated in a humidified incubator at 37 °C supplemented with 5% CO_2_ for up to 72 h. Following a 72 h incubation period, cell culture medium from each well was added to an equal volume of CellTiter-Glo reagent in a white 96-well plate. The plate was mixed on an orbital shaker for 2 min to induce cell lysis followed by a 20 min incubation at room temperature to stabilise the luminescent signal. Luminescence was measured using a Synergy HTX Multi-mode Microplate reader (Biotek, Winooski, VT, USA).

### 4.7. Combination Index Values

Combination index (CI) values were calculated by the Chou and Talalay method [35], using the CompuSyn Software (ComboSyn, Inc., Paramus, NJ, USA). CI values < 1 indicate synergy, CI values = 1 indicate an additive effect and CI > 1 indicates an antagonistic effect.

### 4.8. Western Blot Analysis

Western blot analysis was performed as previously described [61] using antibodies targeting poly-ADP-ribose-polymerase-1 (PARP) (Cell Signalling Technology, Cat # 9542, London, UK), procaspase-3 (Cell Signalling Technology, Cat # 9662, London, UK) and cleaved caspase-3 (Cell Signalling Technology, Cat # 9661, London, UK). Mouse monoclonal antibodies were used in conjunction with anti-mouse IgG, HRP-linked antibody (Cell Signalling Technology, Cat #7076P2, London, UK). Rabbit polyclonal antibodies were used in conjunction with anti-rabbit IgG, HRP-linked antibody (Cell Signalling Technology, Cat # 7074P2, London, UK). Equal loading was assessed using an anti-GAPDH antibody (Cell Signalling Technology, Cat # 97166, London, UK).

### 4.9. Annexin V Flow Cytometry

Cells were seeded at 2 × 10^5^ in a 6-well plate and treated with appropriate drugs. After the 72 h treatment period, the media and cells were transferred to a 15 mL tube on ice and centrifuged at 2000 rpm for 5 min at 4 °C. The supernatant was removed, and the cells were resuspended in 0.5 mL of phosphate-buffered saline (PBS). Following this, a further 3 mL of PBS was added, and the cells were centrifuged at 2000 rpm for 5 min at 4 °C. The supernatant was removed, and the cells were resuspended in 300 µL of 1x binding buffer and transferred to a flow cytometry tube (BD Biosciences, Berkshire, UK). Each sample was subsequently stained with 4 µL of annexin V (BD Biosciences, Berkshire, UK) and 4 µL of propidium iodide (BD Biosciences, Berkshire, UK). The samples were then left to incubate in the dark for 20 min at room temperature. Following the incubation period, 330 µL of 1x binding buffer was added to each sample. Flow cytometry was carried out on a BD FACS Calibur system (BD Bisciences, Berkshire, UK) using the BD CellQuest Pro analysis software (BD Biosciences, Berkshire, UK).

### 4.10. Statistical Analysis

All statistical analysis was performed using the GraphPad Prism 8 software (San Diego, CA, USA). *T*-tests were unpaired and 2-tailed, and 95% confidence intervals were utilised.

## 5. Conclusions

Our in vitro screening strategy has identified a novel therapeutic drug combination for the treatment of paediatric AML. We envision that this approach can be used for a variety of disease-relevant screens allowing the efficient repurposing of drugs that can be rapidly moved into the clinic.

## Figures and Tables

**Figure 1 ijms-22-10163-f001:**
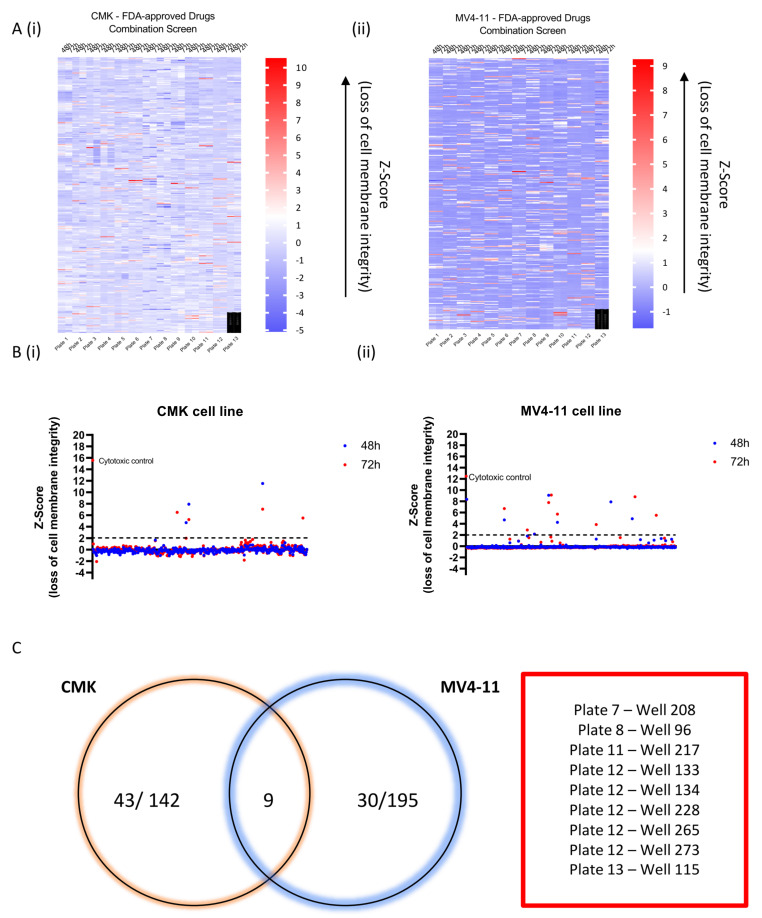
High-throughput screening of paediatric AML cell lines in vitro. All possible pairwise combinations of 384 FDA-approved compounds were assessed (~72,000 wells). An algorithm developed in-house grouped the compounds into pools of 10 compounds per well, reducing the number of wells to 3878. Z-scores were calculated for each of the combinations based on the mean and standard deviation of the normalised relative fluorescence unit (RFU) values. (**A**) Heatmap of the Z-score for each of the 3878 combinations. Time point along the top and the plate number along the bottom. (**i**) CMK; (**ii**) MV4-11. A Z-score > 2 was shown to cause significant loss of membrane integrity correlating with the induction of cell death. (**B**) Graph demonstrating the Z-score of each of the 384 single agents following a 48 and 72 h incubation. (**i**) CMK; (**ii**) MV4-11. (**C**) Venn diagram summarising the successful wells (combination demonstrated a Z-score > 2, while each of the compounds in the well demonstrates minimal single-agent activity) that were common and unique across the cell lines at 72 h.

**Figure 2 ijms-22-10163-f002:**
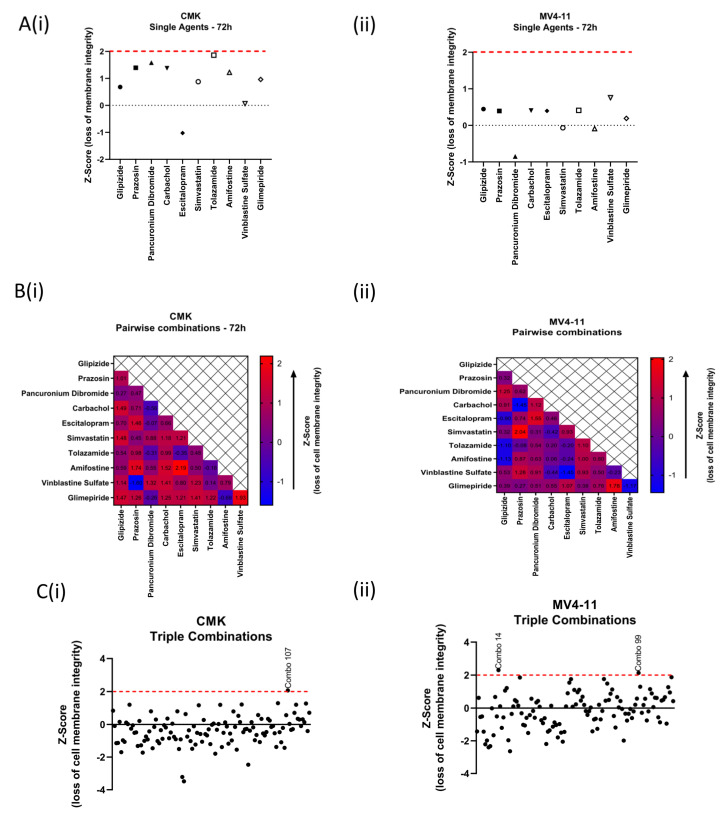
Secondary deconvolution screen of Plate 13 Well 115 into pairwise and triple combinations. The 10 compounds within this well were deconvoluted into 45 pairwise combinations and 120 triple combinations. The viability of these combinations was tested over a period of 72 h at a final concentration of 0.1 µM with the exception of vinblastine sulfate, which was tested at a final concentration of 0.001 µM. (**A**) Graphical representation of the Z-score of each of the 10 compounds as single agents alongside the Z-score of the 10-compound combination. (**i**) CMK; (**ii**) MV4-11. (**B**) Heatmap demonstrating the Z-score of each of the 45 pairwise combinations. (**i**) CMK; (**ii**) MV4-11 (**C**) Graph demonstrating the Z-score of the 120 triple combinations. (**i**) CMK; (**ii**) MV4-11.

**Figure 3 ijms-22-10163-f003:**
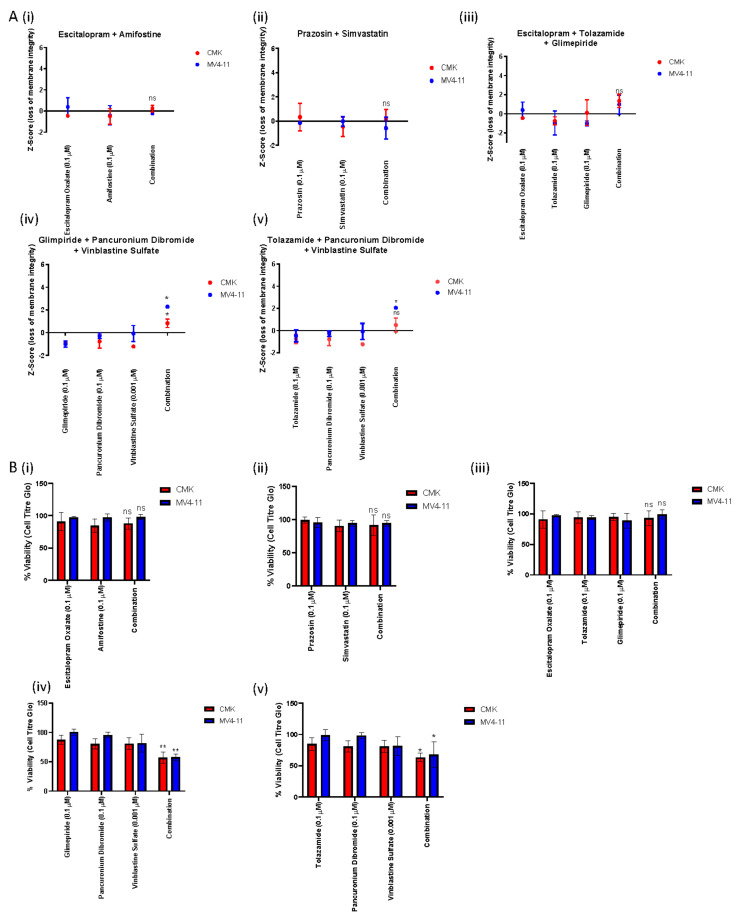
In vitro validation of the pairwise and triple combinations identified in the deconvolution of Plate 13 Well 115. (**A**) Loss of membrane integrity based on Z-score analysis was investigated for each of the pairwise and triple combinations identified in the secondary screen using CellTox green express cytotoxicity assay. One of the combinations was shown to cause a statistically significant increase in Z-score in both cell lines, namely A(**iv**) glimepiride, pancuronium dibromide and vinblastine sulfate (*p* value = 0.017128 and 0.0433105 for the CMK and MV4-11 cell lines, respectively). The combination of A(**v**) tolazamide, pancuronium dibromide and vinblastine sulfate also caused a statistically significant increase in Z-score for the MV4-11 cell line (*p* value = 0.0170). (**B**) Further to this, we investigated the cell viability of each of the combinations identified in the deconvolution using CellTiter-Glo luminescence assay. This assay indicated that two of the combinations identified demonstrated a statistically significant decrease in cell viability compared to the single agents across both cell lines. These were glimepiride, pancuronium dibromide and vinblastine sulfate (*p* value = 0.031 and 0.014 for the CMK And MV4-11 cell lines, respectively) and tolazamide, pancuronium dibromide and vinblastine sulfate (*p* value = 0.0339 and 0.0251 for the CMK And MV4-11 cell lines, respectively). ns = nonsignificant, * = *p* < 0.05, ** = *p* < 0.01.

**Figure 4 ijms-22-10163-f004:**
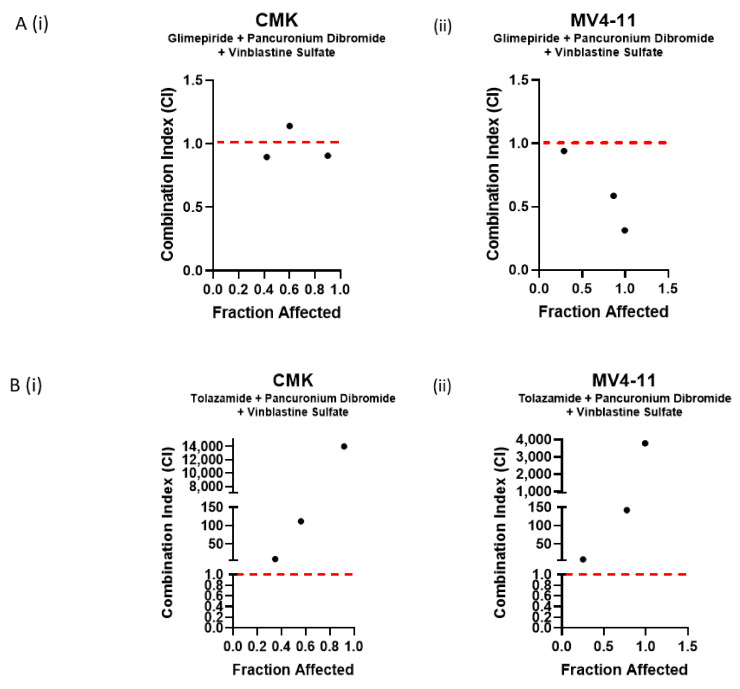
Investigating the synergistic potential of successful combinations using combination index (CI) values. (**A**) Combination index values calculated using the Chou and Talalay method for the combination of glimepiride, pancuronium dibromide and vinblastine sulfate in the (**i**) CMK and (**ii**) MV4-11 cell lines. CI > 1 is antagonistic, CI = 1 is additive and CI < 1 is synergistic. This combination demonstrated synergism at multiple concentrations across both cell lines. (**B**) Combination index values calculated using the Chou and Talalay method for the combination of glimepiride, pancuronium dibromide and vinblastine sulfate in the (**i**) CMK and (**ii**) MV4-11 cell lines. CI > 1 is antagonistic, CI = 1 is additive and CI < 1 is synergistic. This combination demonstrated antagonism at all concentrations across both cell lines.

**Figure 5 ijms-22-10163-f005:**
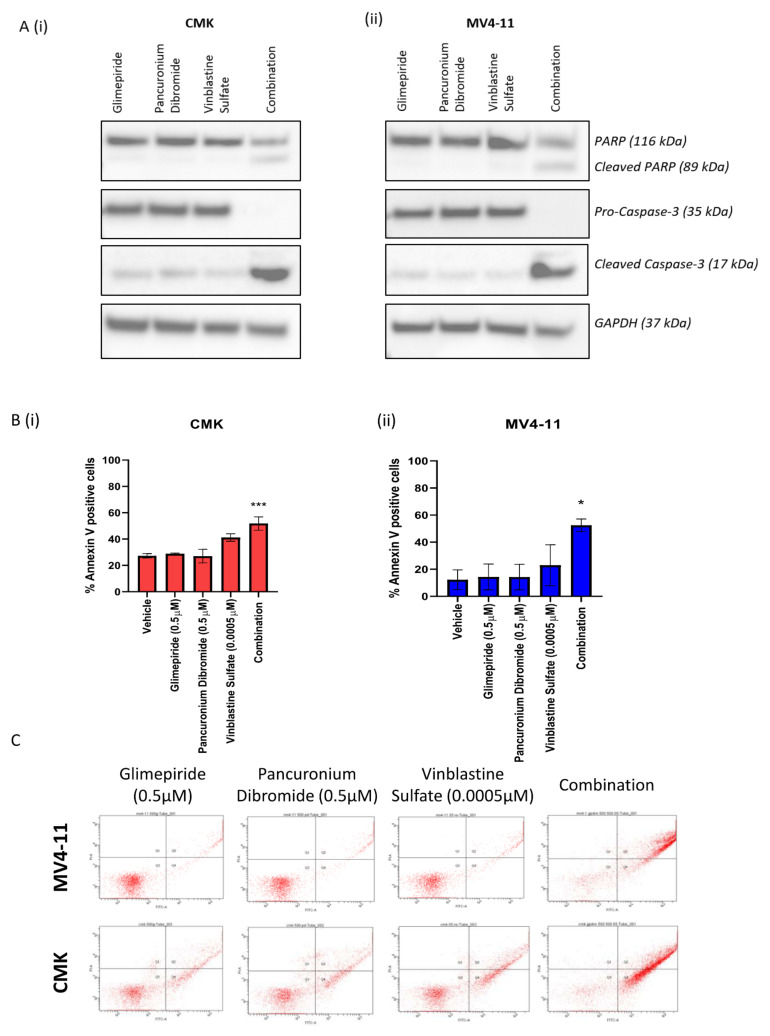
The triple combination of glimepiride, pancuronium dibromide and vinblastine sulfate induces cell death in paediatric AML cell lines. (**A**) Western blot analysis of PARP, procaspase-3, and cleaved caspase-3 in the (**i**) CMK and **(ii**) MV4-11 cell lines following 72 h treatment with glimepiride (0.5 µM), pancuronium dibromide (0.5 µM) and vinblastine sulfate (0.0005 µM) as single agents and as a combination. GAPDH was used as a loading control. Results shown are representative of three independent experiments. (**B**) Flow cytometry analysis of annexin V-positive cell population for the (**i**) MV4-11 and (**ii**) CMK cell lines following 72 h treatment with glimepiride (0.5 µM), pancuronium dibromide (0.5 µM) and vinblastine sulfate (0.0005 µM) as single agents and as a combination. (**C**) Flow cytometry plots of annexin V-positive cell population for the (**i**) MV4-11 and (**ii**) CMK cell lines following 72 h treatment with glimepiride (0.5 µM), pancuronium dibromide (0.5 µM) and vinblastine sulfate (0.0005 µM) as single agents and as a combination. ns = nonsignificant, * = *p* < 0.05, *** = *p* < 0.001.

**Figure 6 ijms-22-10163-f006:**
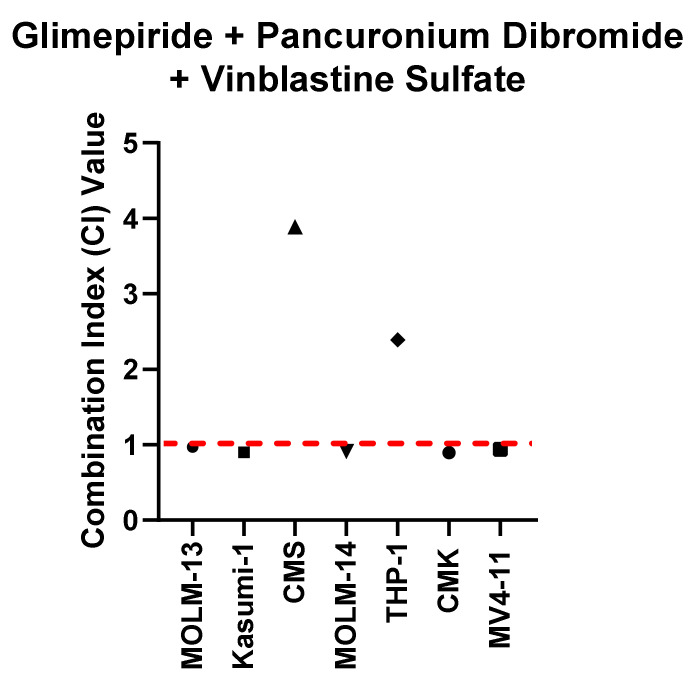
A panel of paediatric AML cell lines were treated with the combination of glimepiride (0.5 µM), pancuronium dibromide (0.5 µM) and vinblastine sulfate (0.0005 µM) for 72 h and combination index values were subsequently calculated using the Chou and Talalay method. CI > 1 is antagonistic, CI = 1 is additive and CI < 1 is synergistic.

**Table 1 ijms-22-10163-t001:** Pairwise and triple combinations that demonstrated a Z-score > 2 in the CMK and MV4-11 cell lines.

CMK		MV4-11	
Pairwise Combinations	Triple Combinations	Pairwise Combinations	Triple Combinations
Escitalopram oxalate + Amifostine	Combo 107 − Escitalopram oxalate + Tolazamide + Glimepiride	Prazosin + Simvastatin	Combo 14 − Glimepiride + Pancuronium Dibromide + Vinblastine Sulfate
			Combo 99 − Tolazamide + Pancuronium Dibromide + Vinblastine Sulfate

**Table 2 ijms-22-10163-t002:** Combination index (CI) values calculated using the Chou and Talalay method for the combination of glimepiride (G), pancuronium dibromide (PD) and vinblastine sulfate (VS).

Glimepiride (µM)	Pancuronium Dibromide (µM)	Vinblastine Sulfate (µM)	CI Value CMK	CI Value MV4-11
0.5	0.5	0.0005	0.89551	0.93907
1	1	0.001	1.14521	0.58593
1	1	0.002	0.90606	0.31275

**Table 3 ijms-22-10163-t003:** Combination index (CI) values calculated using the Chou and Talalay method for the combination of tolazamide (T), pancuronium dibromide (PD) and vinblastine sulfate (VS).

Tolazamide (µM)	Pancuronium Dibromide (µM)	Vinblastine Sulfate (µM)	CI Value CMK	CI Value MV4-11
0.5	0.5	0.0005	9.22723	8.51182
1	1	0.001	119.937	142.212
1	1	0.002	140,007.0	3796.52

## Data Availability

The data presented in this study are available in the article.

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
