# Peer review of "Multiplex Screening for Interacting Compounds in Paediatric Acute Myeloid Leukaemia"

_ijms, 2021, doi:10.3390/ijms221810163_

Round 1

Reviewer 1 Report

In their manuscript, Cairns and collaborators used the in-house designed algorithm MuSICAL to test various combinations of a 384 FDA-compounds library in order to identify combinations of compounds with therapeutic potential to treat pediatric AML. The authors identified a combination of glimepiride, pancuronium dibromide, and vinsblastine sulfate as a potential therapy for pediatric AML and conclude that this type of screening could be used for other malignancies to allow the efficient repurposing of already approved drugs that could be moved rapidly to the clinic.

The manuscript is clear and very well written. Yet, the following comments need to be addressed for the manuscript to be suitable for publication:

Major comments

Since there is no in vivo validation and to strengthen the manuscript, some of the apoptosis assays should at least be performed on the additional cell lines tested in the last paragraph of the result section to validate the mechanism of action of the combination of treatments.

Minor comments

Please correct the term “paediatric” and replace by “pediatric”

The English throughout the manuscript needs moderate improvement. “as previous” needs to be replaced by “as previously described” (line 255) for example. “Furthermore” is better than “further to this” (line 274) etc.

Author Response

Firstly, I would like to thank you for taking the time to read my manuscript and for your useful comments. Below, I have detailed my response to each of your comments:

Major comments  

  1. Reviewer 1 felt that due to lack of in vivo validation some of the apoptosis assays should be performed on the additional cell lines used in Figure 6. Annexin V flow cytometry was performed on the MOLM-13, CMS, Kasumi-1 and MOLM-14 line following 72h treatment with glimepiride, pancuronium dibromide and vinblastine sulfate as single agents and as a triple combination (Supplementary Figure 6A). Alongside this, western blot analysis assessing the expression of PARP, Pro-caspase-3 and Cleaved caspase 3 was performed on the MOLM-13, CMS, Kasumi-1 and MOLM-14 cell line following 72h treatment with glimepiride, pancuronium dibromide and vinblastine sulfate as single agents and as a triple combination (Supplementary Figure 6B). Unfortunately, during this time frame I had difficulty culturing the THP-1 cell line, however, I hope the additional data I have provided will be acceptable without.

Minor Comments

  1. The term ‘paediatric’ has now been replaced with ‘pediatric’ throughout the manuscript.
  2. On line 255 ‘as previous’ has been replaced with ‘as previously described’. This has been changed throughout the manuscript.
  3. On line 274 ‘further to this’ has been replaced with ‘furthermore’. This has been changed throughout the manuscript.

Reviewer 2 Report

In this paper Cairns have conducted screening for interacting compounds in pediatric AML. The scientific rationale is excellent but, still there are real concerns with regards to the manuscript. After reading the paper, I'm not sure why they are holding back critical information such as the heat map of the genes affected and comparison it with the drugs by themselves. This would be critical information in the field especially with this combinatorial drugs. Figure 5 A (i) the control for the combination is not present hence the result is invalid. Figure 5 A (ii) Hard to believe that there is absolutely no cleavage of caspase and then with the combination all are cleaved. Did you observe that all 3 times you repeated the experiment? Since the western blots were repeated 3 times then there should be densitometry histogram data and statistical analysis of that. It is all the more important with GAPDH concentrations  are varying with samples. 

Overall, this is an excellent study and would love to read the resubmission. 

Author Response

Firstly, I would like to thank you for taking the time to read my manuscript and for your useful comments. Below, I have detailed my response to each of your comments:

  1. Reviewer 2 has indicated that we are holding back critical information such as a heat map of the genes affected and comparison of it with the drugs by themselves. We believe you are referring to which genes are altered by the drugs, unfortunatley we do not know the genes which were affected by the individual drugs. Alternatively, you may be referring to a heat map of the cell line mutations and the drug responses, unfortunately, we haven’t done something like that, however, this is something we are hoping to do on a large cohort of mutationally characterised patient samples. We hadn’t reported that in this manuscript as we felt it was perhaps too small a number.
  2. Regarding Figure 5A (i), it appeared that the control (GAPDH) for the combination was missing, however, the blot for GAPDH and Pro-caspase-3 were mixed up, this was an error on my part that has now been corrected. Alongside the manuscript a file containing the original uncropped images was included, in this file the images were labelled correctly.
  3. Regarding Figure 5A (ii), densitometry analysis for these blots was included in supplementary Figure 3. While the image shown in Figure 5A (ii) for cleaved caspase 3 indicates that there is no expression following treatment with the single agents, the densitometry analysis for the three repeats does however show that the expression of cleaved caspase 3 for each of the single agents is variable. Despite this, there is still a statistically significant increase in cleaved caspase 3 for the combination. I have changed the image for cleaved caspase 3 in Figure 5A (ii) to a more representative blot for this cell line. I have also included representative histograms from the densitometry analysis (Supplementary Figure 4).

Reviewer 3 Report

Cairns et al applied a novel screening strategy to identify potential combination therapies for the treatment of paediatric AML. Based on their previously established algorithm designed in-house screening strategy, the current study investigated all combinations of 384 FDA-approved compounds across two paediatric AML cell lines (CMK and MV4-11), which greatly reduced experimental setup with 10-compounds per well.

In general, the study is well designed and presented, the strategy should be considered as approach for other screens to help efficient drug combination identification and clinical application.

I only have some minor comment:

  1. Section 4.3, a final drug concentration of 0.1 µM was used, based on what data/fact was this decided? The dose response curves in Supplementary Figure 2 showed IC50 at lower than 0.001 µM.
  2. Still in Section 4.3, why compounds were diluted with DMSO, is DMSO toxic for cells? All compounds were provided in DMSO for -80°C storage, but DMSO should not be used for dilution, maybe mistake here?
  3. Cell lines used in Figure 6 are not described in Materials and Methods, 4.1. Cell lines
  4. Line 507, ‘Dang et al. (2019) highlighted the potential to improve treatment’, ‘Dang’ is not in reference.

Author Response

Firstly, I would like to thank you for taking the time to read my manuscript and for your useful comments. Below, I have detailed my response to each of your comments:

  1. The final drug concentration of 0.1µM was chosen based on previous work in these cell lines by Lappin et al. (Reference 8). This is the concentration used for the original combination screen however, during the validation process the concentrations were altered to reflect a more clinically relevant situation based on literature and dose response curves (Supplementary figure 2). The dose response curves demonstrate that the IC50 concentration for vinblastine sulfate was 0.003µM and 0.002µM for the CMK and MV4-11 cell lines, respectively. Therefore a concentration of 0.001µM was used as this was below both IC50 concentrations.
  2. This was a mistake, the drugs were originally provided in DMSO, however, during the screening process they were diluted in blank media. This has been rectified in the manuscript.
  3. The cell lines used in Figure 6 were not mentioned in the materials and methods section 4.1, this has been rectified in the manuscript.
  4. Line 507, ‘Dang et al. (2019) highlighted the potential to improve treatment’, ‘Dang’ is not in reference. This was a mistake, the line should read ‘Drenberg et al. (2019)’ to represent the first author of the paper. This is from reference 45. This has been rectified in the manuscript.

Round 2

Reviewer 2 Report

The authors have satisfactorily answered all my queries. They have accepted and addressed some of the critical shortcomings in the manuscript. The manuscript is now ready to be accepted. Congratulations!

Reviewer 3 Report

All issues from my side have been resolved.